# Deep State Space Models for Time Series Forecasting

**Syama Sundar Rangapuram**      **Matthias Seeger**      **Jan Gasthaus**      **Lorenzo Stella**

**Yuyang Wang**                          **Tim Januschowski**

**Amazon Research**
{rangapur, matthis, gasthaus, stellalo, yuyawang, tjnsch}@amazon.com

## Abstract

We present a novel approach to probabilistic time series forecasting that combines state space models with deep learning. By parametrizing a per-time-series linear state space model with a jointly-learned recurrent neural network, our method retains desired properties of state space models such as data efficiency and interpretability, while making use of the ability to learn complex patterns from raw data offered by deep learning approaches. Our method scales gracefully from regimes where little training data is available to regimes where data from large collection of time series can be leveraged to learn accurate models. We provide qualitative as well as quantitative results with the proposed method, showing that it compares favorably to the state-of-the-art.

## 1   Introduction

Time series forecasting is a key component in many industrial and business decision processes. A typical example of such tasks is demand forecasting: accurate and up-to-date models are fundamental to successful inventory planning and minimization of operational costs.

State space models [8, 13, 23] (SSMs) provide a principled framework for modeling and learning time series patterns such as trend and seasonality. Prominent examples include ARIMA models [4, 8] and exponential smoothing [13]. SSMs are particularly well-suited for applications where the structure of the time series is well-understood, as they allow for the incorporation of structural assumptions into the model. This allows for the model to be interpretable and the associated learning procedure to be data efficient, but it requires time series with enough history. In modern forecasting applications with large and diverse time series corpora, SSMs require prohibitively labor- and compute-intensive tasks such as model and covariate selection. Further, traditional SSMs cannot infer shared patterns from a dataset of similar time series, as they are fitted on each time series separately. This makes creating forecasts for time series with little or no history challenging.

Deep neural networks [12, 25, 26] offer an alternative. With their capability to extract higher order features, they can identify complex patterns within and across time series, and can do so from datasets of raw time series with considerably less human effort [9, 27, 19]. However, as these models make fewer structural assumptions, they typically require larger training datasets to learn accurate models. Additionally, these models are difficult to interpret and—often more importantly—make it difficult to enforce assumptions such as temporal smoothness.

In this paper we propose to bridge the gap between these two approaches by fusing SSMs with deep (recurrent) neural networks. We present a forecasting method that parametrizes a particular linear SSM using a recurrent neural network (RNN). The parameters of the RNN are learned jointly from a dataset of raw time series and associated covariates, allowing the model to automatically extract features and learn complex temporal patterns. At the same time, as each individual time series is modeled using an SSM, we can enforce and exploit assumptions such as temporal smoothness.

Furthermore our method is interpretable, in the sense that the SSM parameters for each individual time series can be inspected (and even changed if necessary). By incorporating prior structural assumptions, the presented method scales from small to large data regimes seamlessly. When there is little data to learn from, the structure imposed by the SSM can alleviate overfitting.

The rest of the paper is organized as follows. We first discuss related work in Section 2 and then review the general state space approach to time series forecasting in Section 3. In Section 4, we present our joint forecasting model and describe the training and inference procedure. In Section 5, we first do a qualitative analysis of our method and then present quantitative comparison against the state-of-the-art. We conclude in Section 6.

## 2 Related work

Hyndman et al. [13] and Durbin and Koopman [8] provide comprehensive overviews of SSMs. Recent work in the machine learning literature on linear state-space models includes [23, 22]. We follow [13] in their approach to use linear state space models. The assumption of linear dynamics consisting of interpretable components (level/trend/seasonality) makes the forecasts robust and understandable. Note that non-linear effects can still be captured via exogenous variables. In the forecasting context, non-linearities are typically associated with interventions such as promotions, so this assumption is practically reasonable.

Combining state-space models (SSM) with RNNs has been proposed before, through either or both of the following: (i) extending the Gaussian emission to complex likelihood models; (ii) making the transition equation non-linear via a multi-layer perceptron (MLP) or interlacing SSM with transition matrices temporally specified by RNNs. The so-called *Deep Markov Model* (DMM) proposed by [18, 17] keeps the Gaussian transition dynamics with mean and covariance matrix parameterized by MLPs. *Stochastic RNNs* [10] explicitly incorporate the deterministic dynamics from RNNs by interlacing them with an SSM while the dynamics of RNNs do not depend on latent variables. Compared to DMM, the difference is the added information from the deterministic state of RNNs. An alternative way to make the transition equation non-linear is to cut the ties between the latent states $l_t$'s and associate them with deterministic states $h_t$ of RNN. In this way, the transition from $l_{t-1}$ to $l_t$ is non-linearly determined by the RNN and the observation model. Chung et al. [7] first proposed such *Variational RNNs*. They were later used in *Latent LSTM Allocation* [29] and *State-Space LSTM* [30]. [15] discusses unsupervised learning of state space models from sequential data.

Arguably the most relevant to our work is [11], which aims to keep the linear Gaussian transition structure intact so that the highly efficient Kalman filter/smoother is applicable. The non-linear behavior is approximated by locally linear transition matrices. The so-called called *Kalman Variational Auto-Encoder* (KVAE) disentangles the observations (emissions) and the latent dynamics (transitions) with VAE. By making the locally linear part outside of the standard inference routine and using a fully factorized Gaussian "decoder," Kalman smoothing can be readily applied. A similar idea appeared in [14] where a recognition network is used to output conjugate graphical model potentials so that efficient structural inference is feasible. Our model differs from [11] in that instead of using an RNN to specify the linear combination of a fixed set of $K$ parameters, we directly use RNNs to output the SSM parameters, eliminating the need to tune additional hyper-parameters.

## 3 Background

The general probabilistic forecasting problem is the following. Let $N$ be a set of univariate time series $\{z_{1:T_i}^{(i)}\}_{i=1}^N$, where $z_{1:T_i}^{(i)} = (z_1^{(i)}, z_2^{(i)}, \ldots, z_{T_i}^{(i)})$ and $z_t^{(i)} \in \mathbb{R}$ denotes the value of the $i$-th time series at time $t$.[1] Further, let $\{\mathbf{x}_{1:T_i+\tau}^{(i)}\}_{i=1}^N$ be a set of associated, time-varying covariate vectors with $\mathbf{x}_t^{(i)} \in \mathbb{R}^D$. Our goal is to produce a set of probabilistic forecasts, i.e. for each $i = 1, \ldots, N$ we are interested in the probability distribution of future trajectories $z_{T_i+1:T_i+\tau}^{(i)}$ given the past:

$$p\left(z_{T_i+1:T_i+\tau}^{(i)} \,\middle|\, z_{1:T_i}^{(i)}, \mathbf{x}_{1:T_i+\tau}^{(i)}; \Phi\right). \tag{1}$$

Here $\Phi$ denotes the set of learnable parameters of the model, which are shared between and learned jointly from all $N$ time series. For any given $i$, we refer to time series $z_{1:T_i}^{(i)}$ as *target* time series, to time range $\{1, 2, \ldots, T_i\}$ as *training range*, and to time $\{T_i + 1, T_i + 2, \ldots, T_i + \tau\}$ as *prediction range*. The time point $T_i + 1$ is referred to as *forecast start time* and $\tau \in \mathbb{N}_{>0}$ is the *forecast horizon*. Note that we assume that the covariate vectors $\mathbf{x}_t^{(i)}$ are given also in the prediction range.

We make the common simplifying assumption that the time series are independent of each other when conditioned on the associated covariates $\mathbf{x}_{1:T_i}^{(i)}$ and the parameters $\Phi$. However, in constrast to many related methods that make this assumption, in our approach the model parameters $\Phi$ are shared between *all* time series. So while this assumption precludes us from modeling correlations between time series, it does *not* mean that the proposed model is not able to share statistical strength between and learn patterns across the different time series, as we are learning the parameters $\Phi$ jointly from *all* time series.

**State Space Models.** SSMs model the temporal structure of the data via a *latent state* $\boldsymbol{l}_t \in \mathbb{R}^L$ that can be used to encode time series components such as level, trend, and seasonality patterns. In the forecasting setting they are typically applied to individual times series (though multivariate exensions with multi-dimensional targets exist). For this reason, we will drop the superscript $i$ from the notation in this section. A general SSM is described by the so-called state-transition equation, defining the stochastic transition dynamics $p(\boldsymbol{l}_t|\boldsymbol{l}_{t-1})$ by which the latent state evolves over time, and an *observation model* specifying the conditional probability $p(z_t|\boldsymbol{l}_t)$ of observations given the latent state.

We consider *linear* state space models where the transition equation takes the form

$$\boldsymbol{l}_t = \boldsymbol{F}_t \boldsymbol{l}_{t-1} + \boldsymbol{g}_t \varepsilon_t, \qquad \varepsilon_t \sim \mathcal{N}(0, 1). \tag{2}$$

Here at time $t$, the latent state $\boldsymbol{l}_{t-1}$ maintains information about level, trend, and seasonality patterns and evolves by way of a deterministic *transition matrix* $\boldsymbol{F}_t$ and a random *innovation* $\boldsymbol{g}_t \varepsilon_t$. The structure of the transition matrix $\boldsymbol{F}_t$ and innovation strength $\boldsymbol{g}_t$ determine which kind of time series patterns are encoded by the latent state $\boldsymbol{l}_t$ (see [13] or [22] for details on possible structures; Appendix A.1 in the long version of the paper contains two example instantiations).

The probabilistic observation model then describes how the observations are generated from the latent state $\boldsymbol{l}_t$. Here we consider a univariate Gaussian observation model of the form

$$z_t = y_t + \sigma_t \epsilon_t, \qquad y_t = \boldsymbol{a}_t^\top \boldsymbol{l}_{t-1} + b_t, \qquad \epsilon_t \sim \mathcal{N}(0, 1), \tag{3}$$

where $\boldsymbol{a}_t \in \mathbb{R}^L$, $\sigma_t \in \mathbb{R}_{>0}$ and $b_t \in \mathbb{R}$ are further (time-varying) parameters of the model. Finally, the initial state $\boldsymbol{l}_0$ is assumed to follow an isotropic Gaussian distribution, $\boldsymbol{l}_0 \sim N(\boldsymbol{\mu}_0, \mathrm{diag}(\boldsymbol{\sigma}_0^2))$.

**Parameter learning.** The state space model is fully specified by the parameters $\Theta_t = (\boldsymbol{\mu}_0, \boldsymbol{\Sigma}_0, \boldsymbol{F}_t, \boldsymbol{g}_t, \boldsymbol{a}_t, b_t, \sigma_t)$, $\forall t > 0$. In the classical setting the dynamics are assumed to be time-invariant, that is $\Theta_t = \Theta$, $\forall t > 0$. One generic way of estimating them is by maximizing the marginal likelihood, i.e., $\Theta_{1:T}^* = \mathrm{argmax}_{\Theta_{1:T}} \, p_{SS}(z_{1:T}|\Theta_{1:T})$, where

$$p_{SS}(z_{1:T}|\Theta_{1:T}) := p(z_1|\Theta_1) \prod_{t=2}^{T} p(z_t|z_{1:t-1}, \Theta_{1:t}) = \int p(\boldsymbol{l}_0) \left[ \prod_{t=1}^{T} p(z_t|\boldsymbol{l}_t) p(\boldsymbol{l}_t|\boldsymbol{l}_{t-1}) \right] \mathrm{d}\boldsymbol{l}_{0:T} \tag{4}$$

denotes the marginal probability of the observations $z_{1:T}$ given the parameters $\Theta$ under the state space model, integrating out the latent state $\boldsymbol{l}_t$. In the linear-Gaussian case considered here, the required integrals are analytically tractable.

Note that in the classical setting, if there is more than one time series, a separate set of parameters $\Theta^{(i)}$ is learned for each time series $z_{1:T_i}^{(i)}$ independently. This has the disadvantage that no information is shared across different time series, making it challenging to apply this approach to time series with limited historical data or high noise levels.

## 4 Deep State Space Models

Instead of learning the state space parameters $\Theta^{(i)}$ for each time series independently, our forecasting model instead learns a globally shared mapping from the covariate vectors $\mathbf{x}_{1:T_i}^{(i)}$ associated with each

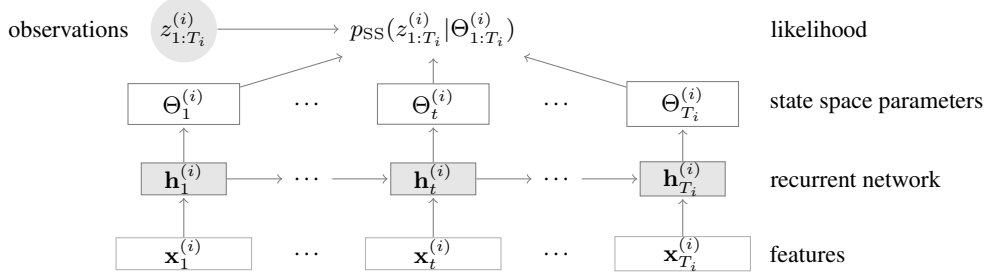

Figure 1: Summary of the model. During training, the inputs to the network are the features $\mathbf{x}_t^{(i)}$ as well as the previous network output $\mathbf{h}_{t-1}^{(i)}$ at each time step $t$ in the training range $\{1, 2, \ldots, T_i\}$. The network output $\mathbf{h}_t^{(i)} = h(\mathbf{h}_{t-1}^{(i)}, \mathbf{x}_t^{(i)}, \Phi)$ is then used to compute the parameters of the state space model $\Theta_t^{(i)}$ after mapping it to the corresponding ranges of the parameters. Given the time series observations $z_{1:T_i}^{(i)}$ in the training range, the likelihood of the state space parameters $\Theta_{1:T_i}^{(i)}$ (which are functions of the shared network parameters $\Phi$) are computed according to Eq. 4. The shared network parameters $\Phi$ are then learned by maximizing the likelihood.

target time series $z_{1:T_i}^{(i)}$, to the (time-varying) parameters $\Theta_t^{(i)}$ of a linear state space model for the $i$-th time series. This mapping,

$$\Theta_t^{(i)} = \Psi(\mathbf{x}_{1:t}^{(i)}, \Phi), \qquad i = 1, \ldots, N, \quad t = 1, \ldots, T_i + \tau, \tag{5}$$

is a function of the entire covariate time series $\mathbf{x}_{1:t}^{(i)}$ up to (and including) time $t$, as well as a set of shared parameters $\Phi$. Then, given the features $\mathbf{x}_{1:T}^{(i)}$ and the parameters $\Phi$, under our model, the data $z_{1:T_i}^{(i)}$ is distributed according to

$$p(z_{1:T_i}^{(i)} | \mathbf{x}_{1:T_i}^{(i)}, \Phi) = p_{SS}(z_{1:T_i}^{(i)} | \Theta_{1:T_i}^{(i)}), \qquad i = 1, \ldots, N. \tag{6}$$

where $p_{SS}$ denotes the marginal likelihood under a linear state space model as defined in Eq. 4, given its (time-varying) parameters $\Theta_t^{(i)}$.

We parameterize the mapping $\Psi$ from covariates to state space model parameters using a deep recurrent neural network (RNN). Figure 1 shows a sketch of the overall model structure, unrolled for all the time steps in the training range. Given the covariates[2] $\mathbf{x}_t^{(i)}$ associated with time series $z_t^{(i)}$, a multi-layer recurrent neural network with LSTM cells and parameters $\Phi$ computes a representation of the features via a recurrent function $h$,

$$\mathbf{h}_t^{(i)} = h(\mathbf{h}_{t-1}^{(i)}, \mathbf{x}_t^{(i)}, \Phi).$$

The real-valued output vector of the last LSTM layer is then mapped to the parameters $\Theta_t^{(i)}$ of the state space model, by applying affine mappings followed by suitable elementwise transformations constraining the parameters to appropriate ranges (see Appendix A.2 in the long version of the paper). Parameters $\Theta_t^{(i)}$ are then used to compute the likelihood of the given observations $z_t^{(i)}$, which is used for learning of the network parameters $\Phi$.

## 4.1 Training

The model parameters $\Phi$ are learned by maximizing the probability of observing the data $\left\{ z_{1:T_i}^{(i)} \right\}_{i=1}^{N}$ in the training range, i.e., by maximizing the (log-)likelihood: $\Phi^{\star} = \operatorname{argmax}_\Phi \mathcal{L}(\Phi)$, where

$$\mathcal{L}(\Phi) = \sum_{i=1}^{N} \log p \left( z_{1:T_i}^{(i)} \middle| \mathbf{x}_{1:T_i}^{(i)}, \Phi \right) = \sum_{i=1}^{N} \log p_{SS} \left( z_{1:T_i}^{(i)} \middle| \Theta_{1:T_i}^{(i)} \right). \tag{7}$$

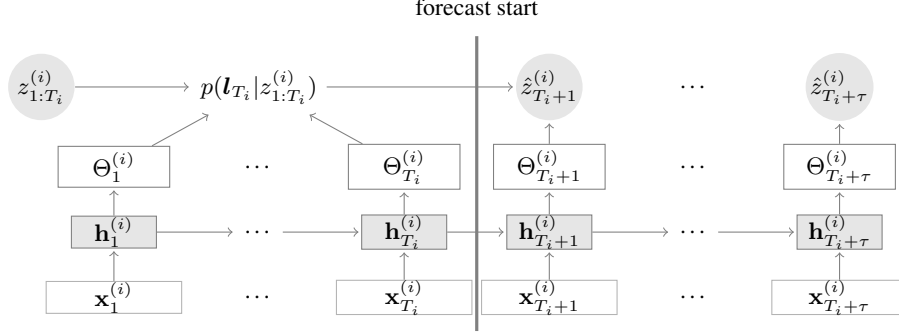

Figure 2: Illustration of the how the model is used to make forecasts after the network parameters $\Phi$ are learned. Given a time series $z_{1:T_i}^{(i)}$ in the training range $\{1, 2, \ldots, T_i\}$ (not necessarily in the training set) and associated features $\mathbf{x}_{1:T_i+\tau}^{(i)}$ for both training and prediction ranges, forecasts are produced as follows: (i) first the posterior of the latent state $p(\boldsymbol{l}_{T_i}|z_{1:T_i})$ for the last time step $T_i$ in the training range is computed using the observations $z_{1:T_i}^{(i)}$ and the state space parameters $\Theta_{1:T_i}^{(i)}$ obtained by unrolling the RNN network in the training range; (ii) given the posterior of the latent state $p(\boldsymbol{l}_{T_i}|z_{1:T_i})$, prediction samples are generated by recursively applying the transition equation and the observation model (Eq. 8) where the state space parameters for the prediction range $\Theta_{T_i+1:T_i+\tau}^{(i)}$ are obtained by unrolling the RNN in the prediction range.

We can view each summand of $\mathcal{L}(\Phi)$ in Eq. 7 as a (negative) loss function, that measures compatibility between the state space model parameters $\Theta_{1:T_i}^{(i)}$ produced by the RNN when given input $\mathbf{x}_{1:T_i}^{(i)}$, and the true observations $z_{1:T_i}^{(i)}$. Each of these terms is a standard likelihood computation under linear-Gaussian state space model, which can be carried out efficiently via Kalman filtering (see e.g. [3, Sec. 24.3] or [22, Appendix A] for details): this involves mainly matrix-matrix and matrix-vector multiplications, which allows us to implement the overall log-likelihood computation using a neural network framework (MXNet), and use automatic differentiation to obtain gradients with respect to the parameters $\Phi$, which are then used by a stochastic gradient descent-based optimization procedure. Note that a forward pass of our network to compute the loss (i.e., negative log-likelihood) essentially uses the same basic primitives as that of classical methods that learn parameters per time series independently. Thus, one can, in principle, extend our ideas to other instances of state space models by simply redefining their parameters as the outputs of the RNN.

## 4.2 Prediction

Once the network parameters $\Phi$ are learned, we can use them to address our original problem specified in Eq. 1, i.e., to make probabilistic forecasts for each given time series. Given $\Phi$, we can compute the joint distribution over the prediction range for each time series analytically, as this joint distribution is a multivariate Gaussian. However, in practice it is often more convenient to represent the forecast distribution in terms of $K$ Monte Carlo samples,

$$\hat{z}_{k,T_i+1:T_i+\tau}^{(i)} \sim p(z_{T_i+1:T_i+\tau}^{(i)}|z_{1:T_i}^{(i)}, \mathbf{x}_{1:T_i+\tau}^{(i)}, \Theta_{1:T_i+\tau}^{(i)}), \qquad k = 1, \ldots, K.$$

In order to generate prediction samples from a state space model, one first computes the posterior of the latent state $p(\boldsymbol{l}_T|z_{1:T})$ for the last time step $T$ in the training range, and then recursively applies the transition equation and the observation model to generate prediction samples. More precisely, starting with sample $\ell_T \sim p(\ell_T|z_{1:T})$, we recursively apply

$$y_{T+t} = \boldsymbol{a}_{T+t}^\top \ell_{T+t-1} + b_{T+t}, \qquad\qquad\qquad t = 1, \ldots \tau, \qquad\qquad (8a)$$

$$\hat{z}_{T+t} = y_{T+t} + \sigma_t \epsilon_t, \qquad \epsilon_{T+t} \sim \mathcal{N}(0, 1), \qquad t = 1, \ldots \tau, \qquad\qquad (8b)$$

$$\boldsymbol{l}_{T+t} \sim \boldsymbol{F}_{T+t}\ell_{T+t-1} + \boldsymbol{g}_{T+t}\varepsilon_{T+t}, \qquad \varepsilon_{T+t} \sim \mathcal{N}(0, 1), \qquad t = 1, \ldots \tau - 1. \qquad (8c)$$

In our case, we compute the posterior $p(\boldsymbol{l}_{T_i}^{(i)}|z_{1:T_i}^{(i)})$ for each of the time series $z_{1:T_i}^{(i)}$ by unrolling the RNN network in the training range to obtain $\Theta_{1:T_i}^{(i)}$ as shown in Figure 2, and then using the Kalman

filtering algorithm. Next, we unroll the RNN for the prediction range $t = T_i + 1, \ldots, T_i + \tau$ and obtain $\Theta_{T_i+1:T_i+\tau}^{(i)}$, then generate the prediction samples by recursively applying above equations $K$ times.[3]

**Remarks.** Note that in our model, in contrast to classical and deep learning-based auto-regressive models (e.g. DeepAR [9]), target values are *not* used as inputs directly. This is a key feature of our method, and brings several advantages: (i) It makes the model more robust to noise, as target values are only incorporated through the likelihood term, where noise is properly accounted for; (ii) Missing target values can easily be handled by simply dropping the corresponding likelihood terms; (iii) Forecast sample path generation is computationally more efficient, as the RNN needs to be unrolled only once for the entire prediction (independent of the number of samples), whereas auto-regressive models (e.g. [9, 26]) have to be unrolled for each sample.

## 5 Experiments

**Qualitative experiments.** In our first experiment, we test whether our model effectively recovers the state space parameters if trained on synthetic data. For this, we generate five groups of time series from day-of-week seasonality model (see Appendix A.1 in the long version of the paper) but with different initial states and innovation parameters per group. For simplicity, we use the same observation noise $\sigma_t$ for all time series. Each time series consists of six weeks of daily data and we use the first four weeks of all time series for training our model. We use a group identifier as an input feature. In the ideal case, for each time series the model should output the parameters of the state space model from which this time series was generated.

The state space model parameters in this case are given by $\Theta_t^{(i)} = (\boldsymbol{\mu}_0^{(i)}, \boldsymbol{\sigma}_0^{(i)}, \gamma_t^{(i)}, \sigma_t^{(i)}), \quad t = 1, \ldots, T_i + \tau$, where $T_i = 28, \forall i$ and $\tau = 14$. Note that except for $\sigma_t^{(i)}$, all the other parameters are different for each group. We encode the day-of-week seasonality using seven components of the latent state as in [22], i.e., $L = 7$ and $\boldsymbol{\mu}_0 \in \mathbb{R}^7$ (each component corresponds to a different day of the week). For simplicity, we fix the term $b_t^{(i)} = 0$ in this experiment.

To analyse how much data is required for recovering the parameters, we train three different models using $N = \{20, 40, 140\}$ examples from each group. Figure 3 shows the ground truth values of the parameters as well as the values obtained by our model for different number of training examples per group. The columns show the mean of the initial state $\boldsymbol{\mu}_0$ (seven values), innovation parameter $\gamma_t$ as well as the standard deviation $\sigma_t$ of the observations while the rows correspond to each of the five groups. The innovation parameter and the standard deviation of the observations are shown for the prediction range (two-weeks). The recovery of state space parameters becomes more accurate gradually as we increase the number of examples from 20 to 140. Moreover, these parameters are recovered reasonably well with $N = 200$ examples per group. In fact, the means of the initial state are exactly recovered. The standard deviation of the initial state $\boldsymbol{\sigma}_0$ (not plotted) has converged to a constant value in all cases. It turns out that the initial state means $\boldsymbol{\mu}_0$ are easy to recover but observation noise $\sigma_t$ and the standard deviation of the initial state $\boldsymbol{\sigma}_0$ are the most difficult to recover.

**Quantitative experiments.** In our first quantitative experiment we evaluate how our model performs under small data regimes. For this, we use the publicly available datasets `electricity` and `traffic` [28]. The `electricity` dataset is a hourly time series of electricity consumption of 370 customers. The `traffic` dataset contains hourly occupancy rates (between 0 and 1) of 963 car lanes of San Francisco bay area freeways. As one expects, all the time series in these datasets exhibit hourly as well as daily seasonal patterns. As baselines we use the classical forecasting methods `auto.arima`, `ets` implemented in R's `forecast` package and a recent RNN-based method DeepAR [9]. We obtained results for DeepAR using the Amazon Sagemaker machine learning platform [1]. Since `DeepAR` and `DeepState` fit a joint model across the time series, both are given a time independent feature representing the category (i.e., the index) of the time series and time-based features like hour-of-the-day, day-of-the-week, day-of-the-month. For `DeepState` the size of SSM (i.e., latent dimension) directly depends on the granularity of the time series which determines the number of seasons. For hourly data, we use hour-of-day (24 seasons) as well as

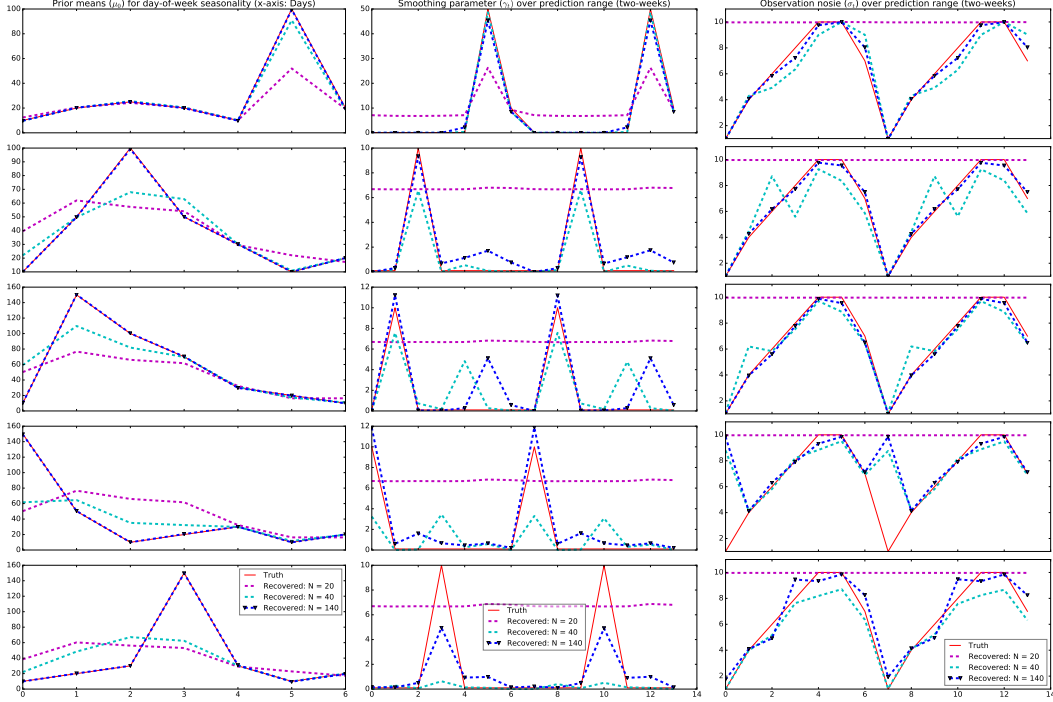

Figure 3: Recovery of state space parameters as the number of examples per group is increased. Columns show state space parameters while the rows correspond to each of the five groups. Each plot shows the true and the recovered values of the parameters with increased number of examples.

| Datasets | Methods | 2-weeks | | 3-weeks | | 4-weeks | |
|---|---|---|---|---|---|---|---|
| | | p50Loss | p90Loss | p50Loss | p90Loss | p50Loss | p90Loss |
| electricity | auto.arima | 0.283 | 0.109 | 0.291 | 0.112 | 0.30 | 0.11 |
| | ets | 0.121 | 0.101 | 0.130 | 0.110 | 0.13 | 0.11 |
| | DeepAR | 0.153 | 0.147 | 0.147 | 0.132 | 0.125 | 0.080 |
| | DeepState | **0.087** | **0.05** | **0.085** | **0.052** | **0.085** | **0.057** |
| traffic | auto.arima | 0.492 | 0.280 | 0.492 | 0.289 | 0.501 | 0.298 |
| | ets | 0.621 | 0.650 | 0.509 | 0.529 | 0.532 | 0.60 |
| | DeepAR | 0.177 | 0.153 | **0.126** | **0.096** | 0.219 | 0.138 |
| | DeepState | **0.168** | **0.117** | 0.170 | 0.113 | **0.168** | **0.114** |

Table 1: Data efficiency. Evaluation on `electricity` and `traffic` datasets with increasing training range. The forecast is evaluated on 7 days.

day-of-week (7 seasons) models and hence latent dimension is 31. We train each method on all time series of these datasets but vary the size of the training range $T_i \in \{14, 21, 28\}$ days. We evaluate all the methods on the next $\tau = 7$ days after the forecast start time using the standard $p50$ and $p90$- quantile losses. For a given collection of time series $z$ and corresponding predictions $\hat{z}$, the $\rho$-quantile loss for $\rho \in (0, 1)$ is defined as

$$\mathrm{QL}_\rho(z, \hat{z}) = 2 \frac{\sum_{i,t} P_\rho(z_t^{(i)}, \hat{z}_t^{(i)})}{\sum_{i,t} |z_t^{(i)}|}, \quad P_\rho(z, \hat{z}) = \begin{cases} \rho(z - \hat{z}) & \text{if } z > \hat{z}, \\ (1 - \rho)(\hat{z} - z) & \text{otherwise.} \end{cases}$$

The $p50$ and $p90$ losses are reported in Table 1. Overall our method achieves the best performance except for one case. Moreover, our method achieves very good performance even with 2-weeks data since it can explicitly incorporate seasonal structures (i.e., hour-of-day seasonality). Although `ets` and `auto.arima` incorporate such seasonal structures, their results are much worse. Inability to learn shared patterns across the time series could be a possible reason for their worse performance. DeepAR tries to learn seasonal patterns purely from the data and its performance generally improves with increased training size. We show some example predictions of our method in Appendix A.5.

Next, to compare against the matrix factorization method [28], we repeat the experiment in [28] that evaluates rolling-day forecasts for seven days (i.e., prediction horizon is one day and forecasts start time is shifted by one day after evaluating the prediction for the current day). Note that unlike MatFact, our method and DeepAR need not retrain after updating the forecast start time. We just extend the training range by one day and update the posterior of the latent state accordingly. The results are shown in Table 2. Since MatFact only produces point forecasts, we report normalized deviation as in [28], which in our case is equal to $p50$-loss. For `DeepAR` and `DeepState` we report both $p50$- and $p90$-losses. Note that our method is much better than MatFact even though the latter is retrained after each day of prediction. We get comparable results to DeepAR, which in our experience performs well with short forecast horizons.

|  | MatFact | DeepAR | DeepState |
|---|---|---|---|
| electricity | 0.16 | **0.075/0.04** | 0.083/0.056 |
| traffic | 0.20 | **0.161/0.099** | 0.167/0.113 |

Table 2: Average $p50/p90$-loss for rolling-day prediction for seven days. MatFact outputs points predictions, so we only report $p50$-loss.

In the final experiment we evaluate our method on a diverse collection of publicly available datasets. We selected datasets containing time series from a single domain as our method is most suited for datasets of related time series. This includes monthly and quarterly time series from the tourism competition dataset [2] describing tourism demand, hourly time series from the M4 competition [20] and `parts` dataset [6] which contains monthly demand of spare parts at a US auto-mobile company. The number of time series in these data sets are 414 (`M4-Hourly`), 366 (`tourism-Monthly`), 427 (`tourism-Quarterly`) and 1046 (`parts`). For tourism and M4 datasets, train and test splits are already provided. The length of the training time series as well as the starting date differ for the time series in the `M4-Hourly` and tourism datasets. The prediction horizon for these data sets are 48 hours (`M4-Hourly`), 24 months (`tourism-Monthly`) and 8 quarters (`tourism-Quarterly`). For `parts` dataset we use the last 12 months as the prediction range while the training range contains 39 months. For both `tourism-Monthly` and `tourism-Quarterly` we used month-of-year seasonal model along with a trend component (to accommodate the trend visible in the training range of these time series) and for `parts` we used month-of-year seasonal model. For `M4-Hourly` we used the hour-of-day as well as day-of-week seasonal models. The $p50$ and $p90$ losses are reported in Table 3 for all the methods. These results further show that our method achieves the best performance overall.

|  | ets | auto.arima | DeepAR | DeepState |
|---|---|---|---|---|
| M4-Hourly | 0.054/0.0267 | 0.052/0.0354 | 0.090/0.0304 | **0.044/0.0266** |
| parts | 1.639/1.0086 | 1.6444/1.0664 | **1.273**/1.086 | 1.47/**0.935** |
| tourism-Monthly | **0.093/0.054** | 0.0999/0.058 | 0.107/0.059 | 0.138/0.067 |
| tourism-Quarterly | 0.105/0.055 | 0.1241/0.062 | 0.11/0.062 | **0.098/0.047** |

Table 3: $p50/p90$-losses for datasets obtained from publicly available sources.

## 6 Conclusions

In this paper we propose a new approach to time series forecasting by marrying state space models with deep recurrent neural networks. This combination allows us to explicitly incorporate structural assumptions to handle small data regimes on one hand and learn complex patterns from raw time series data for larger data regimes on the other hand. Our experiments on synthetic data suggest that the model is capable of accurately recovering the parameters of the state space model from which the data is generated. We also showed, on real-world datasets, that the proposed method achieves state-of-the-art performance by comparing it against a recent RNN-based method, a matrix factorization method, as well as classical approaches such as ARIMA and ETS. Under regimes of limited data our method clearly outperforms the other methods by explicitly modelling seasonal structure. Extending our approach to other instances of state space models as well as non-Gaussian likelihoods are some of the directions we are currently pursuing. Some ideas of extending our method to non-Gaussian likelihoods are discussed in Appendix A.4 in the long version of the paper.

## Footnotes

[1]We consider time series where the the time points are equally spaced, but the units are arbitrary (e.g. hours, days, months). Further, the time series do not have to be aligned, i.e., the starting point $t = 1$ can refer to a different absolute time point for different time series $i$.

[2] The covariates (features) can be time dependent (e.g. product price or a set of dummy variables indicating day-of-week) or time independent (e.g., product brand, category etc.).

[3]Note that the sampling procedure is trivially parallelizable over $K$ samples once the parameters $\Theta_{T_i+1:T_i+\tau}^{(i)}$ and the distribution of the final latent state are computed.

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
