[Supplementary Material]

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

## A Appendix

### A.1 Example Instantiations of the State Space Model

For illustration, we describe a simple level-trend model as well a seasonality-based model, listing out all the parameters that need to be learned. In the level-trend model, the latent state $l_t \in \mathbb{R}^2$ has two dimensions, one for representing the level and the other for the slope of the (linear) trend. The model is given by

$$a_t = \begin{bmatrix} 1 \\ 1 \end{bmatrix}, \quad F_t = \begin{bmatrix} 1 & 1 \\ 0 & 1 \end{bmatrix}, \quad g_t = \begin{bmatrix} \alpha_t \\ \beta_t \end{bmatrix},$$

where $\alpha_t > 0$, $\beta_t > 0$. Both the level and slope components evolve over time by adding innovations $\alpha_t \varepsilon_t$ and $\beta_t \varepsilon_t$ respectively, so that $\alpha_t > 0$, $\beta_t > 0$ are the innovation strength for the level and slope respectively. The level at time $t$ is the sum of level at $t - 1$ and slope at $t - 1$ (linear prediction). The initial state prior $P(l_0)$ is given by $l_0 \sim \mathcal{N}(\mu_0, \text{diag}(\sigma_0^2))$. We learn the state space parameters $\alpha_t > 0$, $\beta_t > 0$, $\mu_0 > 0$, $\sigma_0 > 0$ as well as the external contribution $b_t \in \mathbb{R}$ and the observation noise $\sigma_t > 0$. Thus, for a level-trend model, we have

$$\Theta_t^{(i)} = (\alpha_t^{(i)}, \beta_t^{(i)}, \mu_0^{(i)}, \sigma_0^{(i)}, b_t^{(i)}, \sigma_t^{(i)}), \quad t = 1, \ldots, T_i + \tau.$$

Note that these parameters vary for different time series according to $\Theta_t^{(i)} = \Psi(x_{1:t}^{(i)}, \Phi)$ but all of them are parametrized by the common parameters $\Phi$ of the global RNN model.

In case of seasonality-based models, each seasonality pattern can be described by a set of seasonal factors (or seasons) associated with it. For example, in the day-of-week pattern there are seven factors, one for each day of the week. We can represent each factor as a component of the latent state $l_t \in \mathbb{R}^7$ (see [22] for details). Then, for the day-of-week seasonality model, we have

$$a_t = 1_{\{\text{day}(t)=j\}_{j=1}^7}, \quad F_t = I, \quad g_t = \gamma_t a_t,$$

where $I$ is the identity matrix and $a_t$ is an indicator vector specifying *when a factor is used*. The parameters to be learned in this case are

$$\Theta_t^{(i)} = (\gamma_t^{(i)}, \mu_0^{(i)}, \sigma_0^{(i)}, b_t^{(i)}, \sigma_t^{(i)}), \quad t = 1, \ldots, T_i + \tau.$$

### A.2 Encoding of State Space Model Parameters

In order to constrain the real-valued output values of the RNN to the parameter domains of the state space model, we use the following transformations. Denote by $o_t \in \mathbb{R}^H$ the output of the RNN at time $t$. For any state space model parameter $\theta_t$, we first compute the affine transformation $\tilde{\theta}_t = w_\theta^\top o_t + b_\theta$ with separate weights $w_\theta \in \mathbb{R}^H$ and biases $b_\theta \in \mathbb{R}$ for each parameter $\theta$. All of these weights and biases are included in $\Phi$ and learned. We then transform $\tilde{\theta}_t$ to the domain of the parameter by applying:

- for real-valued parameters, e.g. $b_t$: no transformation,
- for positive parameters $\theta > 0$: the softplus function $\theta_t = \log(1 + \exp(\tilde{\theta}_t))$,
- for bounded parameters $\theta \in [a, b]$: a scaled and shifted sigmoid $\theta_t = (b-a)\frac{1}{1+\exp(-\tilde{\theta}_t)} + a$.

In practice, it is often advisable to impose stricter bounds than theoretically required, e.g. enforcing an upper bound on the observation noise variance $\sigma_t$ or a lower bound on the innovation strengths can stabilize the training procedure in the presence of outliers.

### A.3 Likelihood computation

In this section we provide the technical details for computing the likelihood of our model needed for its training. Recall that the log-likelihood of the observations $\{z_i\}_{i=1}^N$, under our model, is given by

$$\sum_{i=1}^N \log p_{SS}(z_{1:T_i}^{(i)} | \Theta_{1:T_i}^{(i)}), \ \Theta_t^{(i)} = \Psi(x_{1:t}^{(i)}, \Phi).$$

Each of these terms here can be further decomposed as

$$p_{SS}(z^{(i)}_{1:T_i}|\Theta^{(i)}_{1:T_i}) = \prod_{t=1}^{T_i} p(z^{(i)}_t|z^{(i)}_{1:t},\Theta^{(i)}_{1:T_i}).$$

The factors here can be computed using the Kalman filtering algorithm [3]. Here filtering refers to finding the distribution $p(l^{(i)}_{t-1}|z^{(i)}_{1:t}), t = 1, \ldots, T_i$ of the latent state given all the observations up to the current time step. These filtered distributions are Gaussians $p(l^{(i)}_{t-1}|z^{(i)}_{1:t}) = \mathcal{N}(l^{(i)}_{t-1}|f^{(i)}_t, S^{(i)}_t)$ in our case. The mean $f^{(i)}_t$ and covariance $S^{(i)}_t$ of these filtered distributions are found using Kalman filtering algorithm. Since in our case observations at each time point are scalars, the updates in Kalman filtering algorithm involve mainly matrix-matrix and matrix-vector multiplications.

Once we have the filtered distributions, the factors in the likelihood for each of the observations $z^{(i)}$ can be computed as

$$p(z^{(i)}_t|z^{(i)}_{1:t-1},\Theta^{(i)}_{1:T_i}) = \mathcal{N}(z^{(i)}_t|\boldsymbol{\mu}^{(i)}_t, \boldsymbol{\Sigma}^{(i)}_t),$$

where

$$\begin{aligned}
\boldsymbol{\mu}^{(i)}_1 &= \boldsymbol{a}^{(i)\top}_1 \boldsymbol{\mu}^{(i)}_0, & \boldsymbol{\Sigma}^{(i)}_1 &= \boldsymbol{a}^{(i)\top}_1 \boldsymbol{\Sigma}^{(i)}_0 \boldsymbol{a}^{(i)}_1 + \sigma^{(i)^2}_1 && t = 1, \\
\boldsymbol{\mu}^{(i)}_t &= \boldsymbol{a}^{(i)\top}_t \boldsymbol{F}^{(i)}_t \boldsymbol{f}^{(i)}_{t-1}, & \boldsymbol{\Sigma}^{(i)}_t &= \boldsymbol{a}^{(i)\top}_t \left(\boldsymbol{F}^{(i)}_t \boldsymbol{S}^{(i)}_t \boldsymbol{F}^{(i)\top}_t + \boldsymbol{g}^{(i)}_t \boldsymbol{g}^{(i)\top}_t\right) \boldsymbol{a}^{(i)}_t + \sigma^{(i)^2}_t, && t > 1.
\end{aligned}$$

## A.4 Non-Gaussian Likelihoods

A common assumption in classical forecasting methods is normally distributed data. When the data deviates from this assumption, a widely-used approach is to apply a (power) transformation to the data to make them well-behaved, e.g.,via the widely used Box-Cox transformation [5].

In one alternative, Seeger et al. [24, 22] propose a Bayesian latent state forecaster, which, instead of using Gaussian likelihood, admits a Poisson and a *multi-stage* likelihood that is tailored for intermittent sales patterns for large inventory. Laplace approximation in particular ensures the scalability of this approach. However, deriving the Laplace approximation for each likelihood function is cumbersome, and more importantly requires careful numerical attention, especially for non-logconcave likelihoods such as negative binomial.

Here, we briefly discuss a possible approach to handle arbitrary likelihood functions based on *Variational Auto-Encoder* [16, 21] (VAE) framework in the lines of [11]. For this, we first extend the emission equation (3) to

$$z_t \sim \text{Pr}(\cdot|u_t), \qquad u_t = y_t + \sigma_t \epsilon_t,$$

where we have both $u_t$ and $l_t$ as latent variables. The general idea of VAE is to use neural networks to parameterize the observation model $\text{Pr}(z_t|u_t)$ and a variational posterior distribution. The parameters of the neural network are learned jointly to maximize a stochastic approximation of the evidence lower bound (ELBO). In the forecasting case, the typical assumption is that the likelihood function is known in advance. Next we discuss about the inference and model learning. Our goal is to optimize the marginal likelihood of $\boldsymbol{z} := z_{1:T}$, which is intractable without Gaussian emission. We therefore optimize a variational lower bound,

$$\log p(\boldsymbol{z}) = \log \int p(\boldsymbol{z},\boldsymbol{u},\boldsymbol{l}) d\boldsymbol{u}\, d\boldsymbol{l} \geqslant \mathbb{E}_{q_\phi(\boldsymbol{u},\boldsymbol{l})} \log \left[\frac{p(\boldsymbol{z},\boldsymbol{u},\boldsymbol{l})}{q_\phi(\boldsymbol{u},\boldsymbol{l})}\right], \tag{9}$$

where $\boldsymbol{u} = u_{1:T}$, $\boldsymbol{l} = l_{1:T}$ and $q_\phi(\cdot)$ is an NN parameterized by $\phi$ to approximate the variational posterior, also known as the *recognition network*. We use the following variational posterior to approximate the true posterior $p(\boldsymbol{u},\boldsymbol{l}|\boldsymbol{z})$,

$$q_\phi(\boldsymbol{u},\boldsymbol{l}|\boldsymbol{z}) = q_\phi(\boldsymbol{u}|\boldsymbol{z}) p(\boldsymbol{l}|\boldsymbol{u}),$$

where the key is to make the second term coming from the exact conditional posterior from Linear Dynamical System (LDS) such that given $\boldsymbol{u}$, the inference becomes the standard Kalman filtering (see also [11]). Thus, the variational lower bound can be simplified to

$$\begin{aligned}
\mathbb{E}_{q_\phi(\boldsymbol{u},\boldsymbol{l})} \log \left[\frac{p(\boldsymbol{z},\boldsymbol{u},\boldsymbol{l})}{q_\phi(\boldsymbol{u},\boldsymbol{l})}\right] &= \mathbb{E}_{q_\phi(\boldsymbol{u})} \left(\log \left[\frac{p(\boldsymbol{z}|\boldsymbol{u})}{q_\phi(\boldsymbol{u})}\right] + \log p(\boldsymbol{u})\right) \\
&\approx \frac{1}{L} \left(\log \left[\frac{p(\boldsymbol{z}|\widetilde{\boldsymbol{u}}_j)}{q_\phi(\widetilde{\boldsymbol{u}}_j)}\right] + \log p(\widetilde{\boldsymbol{u}}_i)\right), \qquad \widetilde{\boldsymbol{u}}_j \sim q_\phi(\boldsymbol{u}), \quad j = 1, \cdots, L.
\end{aligned}$$

The first term gives rise to the likelihood of the observations. For some cases (e.g., Poisson), we can analytically integrate it out. The second term is the marginal likelihood that can be computed with Kalman Filtering. Different NN structures can be selected for the variational encoder $q_\phi(\boldsymbol{u}|\boldsymbol{z})$. In particular, we consider the bi-directional LSTM to be the desired structure, given that it considers information from both the past and the future, analogously to *backwards messages* in Kalman smoothing. Note that the recognition network is *only* used in the training phrase to better approximate the posterior distribution of the latent Poisson rate (intensity function). Thus, there is no information leak for the desired forecasting use case.

## A.5   Visualization of Forecasts

Here we show example predictions of our model by plotting the median, $p10$ and $p90$ forecasts for two randomly selected time series from `electricity` and `traffic` datasets.

Figure 4: $80\%$ prediction intervals obtained by our method on two randomly selected time series from `electricity` dataset. The training range is two weeks while the prediction range is one week. The forecasts start date is indicated by the green vertical line.

Figure 5: $80\%$ prediction intervals obtained by our method on two randomly selected time series from `traffic` dataset. The training range is two weeks while the prediction range is one week. The forecasts start date is indicated by the green vertical line.