[Reviews · NeurIPS 2018]

Reviewer 1



After reading the author rebuttal: I am pleased with the new experiments, and have increased my score accoedingly. ------------------------------------------------------------------------------- This paper proposes an architecture that uses an RNN to parameterize the matrices that define a linear state space model with Gaussian noise. The model can be trained end-to-end on multiple related time series. Exact inference in the resulting SSM for each individual time series can be done using Kalman filtering. This is a well written paper, that clearly motivates and introduces the proposed deep state space model. As the authors discuss, this paper is closely related to the KVAE model of [9], but instead of using the RNN to compute the weights of a mixture of globally learned SSM matrices, the RNN is now used to define the matrices themselves. Time-series forecasting is however a novel application for this type of models, and I found it interesting to see that this combination of deep learning and probabilistic modelling techniques works well in the experiments. While there are some initial promising results, I would have expected a stronger experimental section to support your claims, with additional experiments in the large-scale setting and possibly with non-gaussian likelihoods. Some important experimental details are missing. For example, how big are the RNNs/SSMs in each of the experiments? Which are the covariates in the electricity and traffic experiments? The authors claim that by incorporating prior structural assumptions the model becomes more interpretable. Wouldn't the RNN learn a complex non-linear mapping of the input that makes it difficult to understand how the inputs to the model affect the final prediction? A related work you did not cite is: Karl et al. Deep Variational Bayes Filters: Unsupervised Learning of State Space Models from Raw Data. ICLR 2017 Also, citations [8] and [9] are duplicates.

Reviewer 2



This paper proposes introducing deep neural networks to output the time-varying parameters of a linear state-space model (SSM), preserving the analytical tractability of SSM (e.g. Kalman filtering equations, in the case of a Gaussian likelihood) with increased flexibility coming from time-varying SSM parameters. In the paper, an LSTM maps from the covariates to the set of SSM parameters, and the LSTM captures time-series effects in the parameters. Overall, the paper is very well written and quite easy to follow. The review of SSM is adequate, and the connection with the proposed approach is well made and intuitive. As to the technical contribution itself, the DeepState method of incorporating deep neural networks within a SSM appears novel, and worthy of study within the important topic of time series forecasting. Important connections to the recent literature, such as Fraccaro et al. (2017), Johnson et al (2016) and Krishnan et al. (2017) are provided, and helpful. I have three main questions & comments for this review. First, as stated on lines 125-130, the main use of the LSTM is to non-linearly map from covariates to SSM parameters (sharing this mapping across all time series). An obvious question is why are past observations of a given time series not also provided as adjunct to the covariates, not unlike the treatment within DeepAR. Are there identifiability issues to doing so? This should merit some discussion in the paper. Second, the experimental evaluation first seeks to recover known generative parameters on synthetic data, which it appears to successfully carry out. Sadly, the evaluation on real datasets is somewhat disappointing, with a comparison against only two multi-series datasets. The evaluation metric is limited to the quantile loss, which is considers all time-steps over the prediction horizon to be independent of one another — whereas the proposed method (as stated on line 83) aims at making predictions over complete future trajectories. It would have been nice to report some measure of the predictive likelihood of this distribution over complete trajectories (using, e.g., some form of kernel density estimation to smooth over the Monte Carlo predictions). Moreover, the loss function QL is a normalized variant of the well-known pinball loss, but a citation would nonetheless be in order. Separately, the set of comparison models leaves to be desired and does not allow appreciating the contribution to performance of the different model components. It is not clear whether the auto.arima and ets model have access to the covariates that DeepState and DeepAR have. Since DeepState extends well-known SSMs, it should test against two simple baselines: (1) a standard fixed-parameter Linear Dynamical System (LDS) that does not use covariates, and (2) one that does use them linearly. Moreover, it would be helpful to run a modicum of ablation studies, e.g. removing the recurrence in the neural network, and varying the dimensionality of the hidden state h_t (which does not appear to be reported). Third and finally, the section 4.3 is interesting and relevant, but seems to have been hastily written and does not really fit within the rest of the manuscript. In particular, none of the experiments make use of the approach, which would be best suitable in contexts where the time series consist of non-negative observations, with possibly fat tails. Overall, I think the paper proposes a fruitful line of inquiry, but remains in too early a state to recommend acceptance at NIPS. [Update: I think the authors' rebuttal was helpful, including running additional experiments, and with the proposed changes I have updated by score to a 6.] Detailed comments: * Line 52: efffects ==> effects * Line 61: dependent ==> depend * Line 77: acronym LDS (presumably Linear Dynamical System) is not defined * Line 88: are the covariates always assumed to be non-stochastic, even over the prediction range? * Eq. (2): it’s not clear why the innovation \epsilon_t is assumed to be a scalar, which forces the random disturbance to the latent state to be strictly along g_t. In many state-space models, the transition dynamics is assumed to have noise dimension to be equal to the latent space dimension, so this discrepancy should be explained. * Line 114: the dynamics is ==> the dynamics are * Footnote 3: the the ==> the * Equation 8: should be more careful about notation, in particular indicating that the integration is carried out wrt variables u and l. * Line 238-240: notation overload with \tau used to denote forecast horizon and evaluation quantile. * References [8] and [9] are identical. * Add to to background work on linear state space models, this paper of relevance: Chapados, N., 2014. Effective Bayesian Modeling of Groups of Related Count Time Series. In International Conference on Machine Learning (pp. 1395-1403).

Reviewer 3



This paper tackles the problem of forecasting -- predict into a fixed horizon the future of a univariate random variable given its history and the forecast for the associated covariates. In this approach, instead of learning the (time varying) state space parameters of each time series independently, the forecasting model maps from covariate vectors to the parameters of a linear SSM for each time series using an RNN. The intuition is that the covariates contain sufficient information that an RNN can learn to predict the parameters corresponding to the linear time-varying dynamical system. Experimentally, the paper tests the sample complexity of the method on synthetic data. On two publicly available datasets, the proposed method outperforms a baseline of DeepAR and ARIMA, particularly in the small data regime where there is only a short time horizon that is provided to condition upon. Overall, the paper is well written and easy to follow. The idea is simple and appears, from the presented results, effective. As one might expect, this kind of model works best when the forecasts are being performed using very short initial time-horizons. One concern that I have is the choice to include Section 4.3 in this paper. I understand the idea behind it (which is that for non-Gaussian likelihoods, one can use VAEs to maximize the likelihood of data and use variational learning for parameter estimation). However, (from what I understand) it is not experimented with and is primarily proposed as a potential extension. My suggestion would be to explore this idea in a different submission. In its current form, it only serves to show that the current proposal has potential future work.